# Spontaneous Bacterial Peritonitis: A Rare Complication of Pulmonary Arterial Hypertension

**DOI:** 10.3390/biomedicines12071389

**Published:** 2024-06-22

**Authors:** Taylor Beckmann, Nicholas Pavlatos, Dinesh K. Kalra

**Affiliations:** 1Department of Internal Medicine, University of Louisville School of Medicine, Louisville, KY 40202, USA; taylor.beckmann@louisville.edu (T.B.); nicholas.pavlatos@louisville.edu (N.P.); 2Division of Cardiology, University of Louisville School of Medicine, Louisville, KY 40202, USA

**Keywords:** pulmonary hypertension, right-sided catheterization, pulmonary circulation, right ventricle, echocardiography

## Abstract

Approximately 3% of all diagnosed cases of ascites are of cardiac etiology. Although more commonly associated with heart failure, pulmonary arterial hypertension is a known but rare cause of cardiac ascites, which has not been associated with spontaneous bacterial peritonitis. We present a case of a 75-year-old male with known pulmonary arterial hypertension and new-onset ascites, the fluid analysis of which was consistent with both cardiac ascites and spontaneous bacterial peritonitis. He was successfully managed with antibiotics, loop diuretics, and mineralocorticoid receptor antagonists.

## 1. Introduction

Ascites is defined as an abnormal fluid collection within the peritoneal cavity. Patients typically present with abdominal distension/discomfort, dyspnea, early satiety, and weight gain. Depending on the etiology, ascites may develop over the course of days or months. The majority of ascites cases are a result of liver cirrhosis (81%), with other common causes being malignancy (10%), cardiac (3%), and tuberculosis (2%) [1]. Cardiac ascites may arise from right heart failure, constriction, or obstruction to right heart flow. Typically, these conditions result in an increase in right atrial pressure which is transmitted through the inferior vena cava to the hepatic venous system and, ultimately, the sinusoids of the liver, resulting in portal hypertension. Sinusoidal congestion leads to a protein-rich exudate that is initially drained by the liver’s lymphatic system. Once the volume of exudate overwhelms the lymphatic system, it collects in the peritoneal cavity, resulting in ascites [2].

Spontaneous bacterial peritonitis (SBP) is an ascitic fluid infection without an evident intra-abdominal, surgically treatable source [3]. It is mostly associated with ascites secondary to cirrhosis; however, in 1984, the first reported case of SBP in the setting of cardiac ascites was published. Since then, few other case reports have been published, all of which have been in the setting of heart failure [4,5,6]. The scarcity of SBP in congestive hepatopathy is proposed to be due to preserved opsonic and bactericidal activity of the high protein ascitic fluid as well as maintained integrity of the hepatic reticuloendothelial system, unlike in ascites secondary to advanced cirrhosis [7]. We present a rare case of SBP in the setting of ascites as a result of pulmonary arterial hypertension (PAH).

## 2. Patient Case

A 75-year-old male presented to the emergency department with a two-month history of progressively worsening lightheadedness, dyspnea, and abdominal swelling. He denied excessive alcohol use, recent abdominal pain, vomiting, jaundice, rectal bleeding, and a history of liver dysfunction. Past medical history was remarkable for type 2 diabetes mellitus and WHO Group 1 PAH, which had been diagnosed 8 months previously by right heart catheterization. At that time, the patient had an increased mean pulmonary artery pressure of 34 mmHg (while on drug therapy for PAH with macitentan 10 mg daily and sildenafil 20 mg three times daily), normal pulmonary capillary wedge pressure of 8 mmHg, increased pulmonary vascular resistance of 932.25 dynes/sec/cm^−5^ or 11.6 Wood Units, decreased cardiac output of 3.82 L/min, decreased cardiac index of 1.73 L/min/m^2^, and a Registry to Evaluate Early and Long-term PAH Disease Management (REVEAL) score of 10. On transthoracic echocardiogram (TTE), his ejection fraction (EF) was 65%, right atrial volume was 137 mL, right ventricular end-diastolic diameter was 4.7 cm, and right ventricular systolic pressure was 80 mmHg, consistent with severe pulmonary hypertension without left ventricular dysfunction. His right ventricular systolic function was noted to be severely reduced. The ventilation–perfusion scan was negative for the acute or chronic pulmonary embolic disease. The patient was thought to have WHO Group 1 PAH as both his pulmonary artery pressure and pulmonary vascular resistance were elevated, while his pulmonary capillary wedge pressure was normal. Additionally, he had no history of any other causes of pulmonary hypertension such as left heart failure, pulmonary disease, recurrent pulmonary embolism, sarcoidosis, or metabolic diseases.

Upon arrival at the emergency department, his blood pressure was 84/64 mmHg, which responded to 500 mL intravenous normal saline. While on his baseline of 4 L supplemental oxygen, his oxygen saturation was 89%. Physical examination was notable for significant respiratory distress with bilateral expiratory wheezing, jugular venous distension when positioned at 45°, and abdominal distension with a positive fluid wave. A loud P2 and holosystolic murmur increased with inspiration on cardiac auscultation. His lower extremities were noted to have 2+ peripheral edema to the knees bilaterally.

Notable normal admitting lab values included the following: aspartate aminotransferase, alanine transaminase, alkaline phosphatase, bilirubin, platelet count, and international normalized ratio. Creatinine on arrival was 0.79 mg/dL and white blood cell count was 6.0 cells/μL. The only noteworthy abnormal lab was an elevated brain natriuretic peptide of 864 pg/mL. Venous blood gas on arrival with pH of 7.353, pCO_2_ 55.9 mmHg, HCO_3_ 31.1 mmHg, and pO_2_ 21 mmHg. Large-volume paracentesis was performed (5 L), with fluid analysis revealing a serum–ascites albumin gradient (SAAG) of 1.2 g/dL and total protein of 2.6 g/dL. The total nucleated cell count was 1020 cells/mm^3^ with a polymorphic nucleated cell count (PMN) of 663 cells/mm^3^, consistent with SBP. The bacterial culture of peritoneal fluid did not grow any organism. An abdominal ultrasound revealed nodular contours within the liver and heterogeneous echotexture compatible with cirrhosis (Figure 1). Doppler imaging of the liver showed patent vessels. Repeat TTE on this admission showed a preserved EF of 65% with paradoxical septal motion, flattened interventricular septum, and severe right ventricular (RV) enlargement consistent with RV pressure/volume overload. The pulmonary systolic pressure was estimated at 74 mmHg (Figure 2). Additionally, the patient was found to have moderate tricuspid regurgitation with a maximum velocity of 385 cm/second (Figure 3).

Following paracentesis, the patient was transfused 50 g of 25% albumin per American Association for the Study of Liver Disease (AASLD) guidelines. Upon results of fluid analysis indicating SBP, he was treated with Ceftriaxone 2 g intravenously daily for seven days, followed by lifelong prophylaxis with oral Ciprofloxacin 500 mg daily. In light of the patient’s signs of volume overload and right heart failure, the decision was made to not follow SBP protocol, which would have called for additional albumin. It is important to note that the guidelines for the management of SBP have been written by the AASLD specifically for SBP in hepatic cirrhosis [8]. There are currently no guidelines on management in ascites from other sources. Although this patient’s abdominal ultrasound showed coarsened echotexture within the liver and ascites, he had no other signs or manifestations consistent with cirrhosis. Given his normal liver synthetic function and Model for End-Stage Liver Disease 3.0 score of 8, these findings on ultrasound were felt to represent structural changes of congestive hepatopathy and not a primary process.

Repeat therapeutic paracentesis was performed on day seven, which showed less than 250 PMNs, indicative of resolution of his infection. Upon discharge, his home diuretic regimen of furosemide 40 mg daily was doubled and spironolactone 25 mg daily was added. Nutrition counseling was performed in which the patient was instructed to adhere to a 2 g/day sodium diet and follow a fluid restriction of less than 2 L/day.

Since discharge, the patient has continued to follow up with the pulmonary hypertension clinic and has had no reaccumulation of ascites or any further hospitalizations. His current medication regimen is macitentan 10 mg daily, tadalafil 40 mg daily, furosemide 60 mg daily, and spironolactone 25 mg daily. In subsequent follow-up appointments, the patient’s international normalized ratio remained persistently elevated to 1.4, suggesting poor synthetic function of the liver. His liver enzymes have remained within normal limits, which is consistent with a chronic cirrhotic process.

## 3. Discussion

Cardiac ascites are estimated to account for ~3% of ascites cases and should be suspected in patients presenting with new-onset ascites and right heart failure [1]. Diagnostic paracentesis to calculate SAAG and measure total protein is required to determine the etiology of any new-onset ascites. A SAAG ≥ 1.1 g/dL is 85% sensitive and 61% specific for portal hypertension as the cause of ascites [9,10]. Cardiac and post-hepatic ascites typically have ≥2.5 g/dL protein since hepatic sinusoids remain permeable in heart failure whereas the fibrosis in cirrhosis prevents leakage of protein out of the liver (Figure 4) [10,11]. This patient’s SAAG was 1.2 g/dL with a total protein level of 2.6 g/dL, consistent with a cardiac or post-hepatic cause of portal hypertension. Doppler ultrasound and computed tomography ruled out hepatic or portal vein occlusion and given the severity of his PAH, TTE findings, and clinical symptoms, it was concluded that decompensated right heart failure was the cause of his ascites. A combined SAAG ≥ 1.1 g/dL and peritoneal fluid total protein ≥ 2.5 g/dL favors heart failure-induced ascites with 54% sensitivity and 88% specificity, and a brain natriuretic peptide > 364 pg/mL favors cardiac ascites with a sensitivity of 98% and specificity of 99% [12,13]. 

SBP is a life-threatening infection of ascitic fluid with no obvious source, such as abdominal perforation, which commonly presents with nonspecific symptoms. This infection is thought to arise from the translocation of gut bacteria into peritoneal fluid due to numerous factors including structural changes in the intestinal mucosa, intestinal bacterial overgrowth, and changes in intestinal immunity. These changes are specific to the pathophysiology of hepatic cirrhosis and are not linked to the pathophysiology of cardiac ascites [7,14]. Ascitic fluid cultures and gram stains are often negative and can take over 24 h to yield results; thus, the diagnosis is based upon the PMN cutoff of 250 cells/mm^3^. With over 650 PMNs, our patient met the diagnostic criteria for SBP and was treated as such.

Ascites is a rare complication in the progression of PAH and occurs in the later stages as right ventricular uncoupling occurs and right heart failure sets in. Right heart failure is a post-hepatic cause of portal hypertension and results in volume overload of the venous system and elevated pressures in the right atrium. Given the lack of valves in the hepatic veins, the elevated pressures in the right atrium are directly transferred backward to the hepatic veins [11,15]. The increased sinusoidal pressure results in cholestasis and biliary injury. Reactive inflammation from ischemia and injury can lead to fibrosis over time. Fibrosis impairs blood flow through the portal vein leading to elevated portal pressure and portal hypertension. The increased pressure also leads to fluid accumulation and swelling of the liver capsule, causing hepatomegaly and transudation from the hepatic and portal veins into the abdominal cavity causing ascites [15]. This patient’s new development of ascites with corresponding jugular venous distension represents the progression of his PAH into right heart failure. His progression represents treatment failure on dual therapy with a phosphodiesterase-5 inhibitor (sildenafil) and endothelin receptor antagonist (macitentan). He will likely need advancement to triple therapy with prostacyclin analog upon follow-up with his PAH pulmonologist. Spironolactone was also added to his diuretic regimen for its potassium sparring effect.

There are few case reports of SBP in cardiac ascites from various causes of right heart failure, but none could be found of this occurring in PAH [4]. Cardiac ascites are generally considered “low-risk” for peritoneal infections due to protein-rich ascitic fluid where opsonic and bactericidal activity remains intact. It has been theorized that in the setting of left heart failure and poor cardiac output, chronic intestinal ischemia and reperfusion can lower local pH and act as virulence factors for local microorganisms [15]. Although our patient’s EF was within normal limits, it is possible that venous congestion within the intestinal microvasculature from portal hypertension could have caused ischemic damage leading to a similar effect.

## 4. Conclusions

This patient presented with new-onset ascites and SBP. His ascites were determined to be cardiac in origin, resulting from congestive hepatopathy given the following: SAAG ≥ 1.1 g/dL, ascitic fluid total ≥ 2.5 g/dL, brain natriuretic peptide > 364 pg/dL, intact liver synthetic function, preexisting severe PAH, and physical signs of right heart failure. Given his signs of right heart failure, following the AASLD guidelines of SBP management posed too great a risk of intravascular volume overload. There is a need for guidelines on the management of SBP in the setting of cardiac ascites.

## Figures and Tables

**Figure 1 biomedicines-12-01389-f001:**
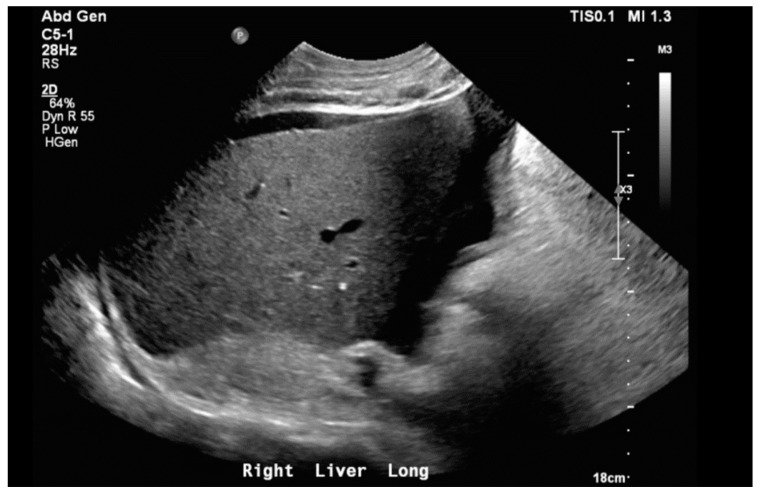
Liver ultrasound in right long axis demonstrating nodular contours with coarsened echotexture. Moderate volume ascites can also be seen.

**Figure 2 biomedicines-12-01389-f002:**
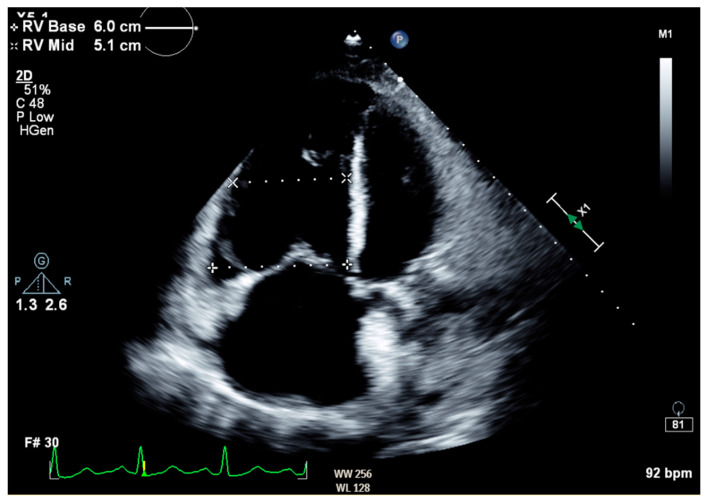
Transthoracic echocardiogram in apical four-chamber view showing severe right atrial and ventricular enlargement, as well as flattened interventricular septum, consistent with elevated RV pressure and volume overload.

**Figure 3 biomedicines-12-01389-f003:**
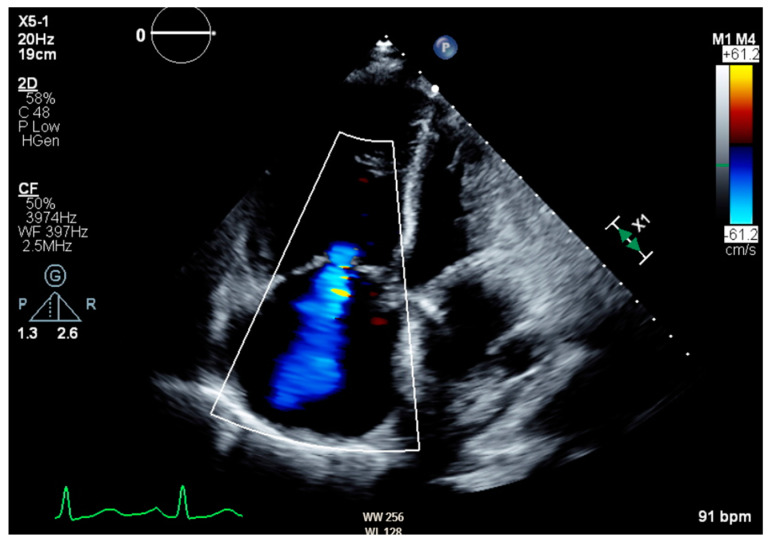
Transthoracic echocardiogram in apical four-chamber view showing moderate tricuspid regurgitation with a maximum velocity of 385 cm/second and max pressure gradient of 59.3 mmHg.

**Figure 4 biomedicines-12-01389-f004:**
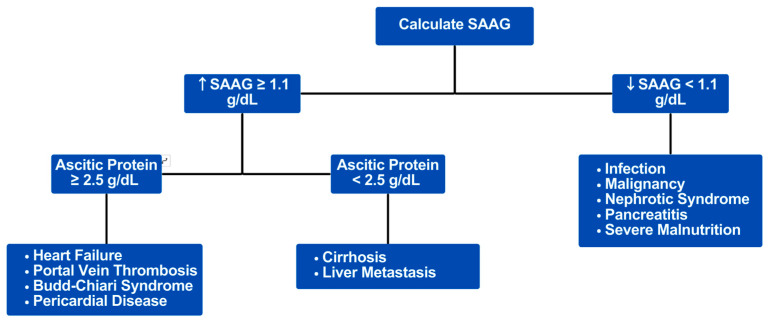
Diagnostic pathway in ascitic fluid analysis: SAAG can be used to differentiate between portal and non-portal hypertension etiologies of ascites. A low SAAG (<1.1) indicates ascites due to non-portal hypertension while an elevated SAAG (≥1.1) favors elevated portal hypertension. Ascitic protein can further help subdivide portal hypertension causes of ascites, where an elevated ascitic protein (≥2.5) indicates cardiac and post-hepatic ascites, whereas a low ascites protein (<2.5) favors cirrhosis.

## Data Availability

The original contributions presented in the study are included in the article, further inquiries can be directed to the corresponding author.

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
