# Peer review of "Spontaneous Bacterial Peritonitis: A Rare Complication of Pulmonary Arterial Hypertension"

_biomedicines, 2024, doi:10.3390/biomedicines12071389_

Round 1

Reviewer 1 Report

Comments and Suggestions for Authors

This is an interesting case report of the spontaneous bacterial peritonitis in patient with cardiac ascites due to pulmonary hypertension caused right heart failure. 

It is unusual to make the diagnosis of PAH type 1 in 75-year old patient. What kind of PH type 1 is presented here? It is also unusual that patient had normal transaminase blood levels and bilirubin. Please present us LDH, gama-GT levels and creatinine blood levels. 

MPAP is low and it is also unusual with ascites caused by the cardiac reason. 

Can we see tricuspide regurgitation and velocity of TR. 

Please comment these observations. 

Reviewer 2 Report

Comments and Suggestions for Authors

In this article entitled “Spontaneous Bacterial Peritonitis: A Rare Complication of Pulmonary Arterial Hypertension” the authors report an exceptional case of SBP, investigating the etiological role of pulmonary hypertension in determining the disease. The case is well presented, reporting the few previously available evidence and addressing the specific aspects of their case. The English style is fine, and the case is interesting. The references are appropriate.

I have few minor comments:

1.     The reported Cardiac Output (3.82 L/min) should be indexed to BSA.

2.     If available, the authors may report the results of the blood gas analysis at admission to ER.

3.     How do the authors explain the absence of peripheral oedema, given the significant right heart failure and also SBP. This may be discussed in section “discussion”.

Comments on the Quality of English Language

English style is fine

Reviewer 3 Report

Comments and Suggestions for Authors

The authors reported a 75-year-old man with known pulmonary arterial hypertension who developed ascites, later found to be both cardiac in origin and complicated by spontaneous bacterial peritonitis. He was effectively treated with antibiotics, loop diuretics, and mineralocorticoid receptor antagonists.

General Comments:

This manuscript addresses "Spontaneous Bacterial Peritonitis: A Rare Complication of Pulmonary Arterial Hypertension." It is well-written; however, there are some concerns that need to be addressed.

Specific Comments:

1.     I doubt that we can refer to this pathology as "congestive hepatopathy" without observing any changes in aspartate aminotransferase, alanine transaminase, alkaline phosphatase, and bilirubin levels.

2.     Line 67: Data on WBC and CRP levels would also be necessary.

3.     The results of the bacterial culture are necessary.

4.     Information regarding the patient's latest status and any recurrence of SBP would be needed.

Round 2

Reviewer 1 Report

Comments and Suggestions for Authors

Authors answered to all comments I have given. I recommend acceptance. 

Reviewer 3 Report

Comments and Suggestions for Authors

The authors have corrected the manuscript according to the reviewer's comments.